



# Map of forest tree species for Poland based on Sentinel-2 data

Ewa Grabska-Szwagrzyk[1], Dirk Tiede[2], Martin Sudmanns[2], Jacek Kozak[1]

[1]Institute of Geography and Spatial Management, Jagiellonian University, Gronostajowa 7, 30-387 Kraków, Poland
[2]Department of Geoinformatics—Z_GIS, University of Salzburg, Schillerstr. 30, 5020 Salzburg, Austria

*Correspondence to*: Ewa Grabska-Szwagrzyk, ewa2.grabska@uj.edu.pl

**Abstract**

Accurate information on forest tree species composition is vital for various scientific applications, as well as for forest inventory and management purposes. Country-wide, detailed species maps are a valuable resource for environmental management, conservation, research, and planning. Here, we performed the classification of 16 dominant tree species/genera

in Poland using time series of Sentinel-2 imagery. To generate comprehensive spectral-temporal information, we created Sentinel-2 seasonal aggregations known as Spectral-Temporal Metrics (STMs) within Google Earth Engine (GEE). STMs were computed for short periods of 15-30 days during spring, summer, and autumn, covering multi-annual observations from years 2018 to 2021. The *Polish Forest Data Bank* served as reference data, and, to obtain robust samples with pure stands only, it was validated through automated and visual inspection based on very high resolution orthoimagery, resulting in 4500

polygons, serving as training and test data. The forest mask was derived from available land cover datasets in GEE, namely ESA World Cover and Dynamic World. Additionally, we incorporated various topographic and climatic variables from GEE to enhance classification accuracy. The Random Forest algorithm was employed for the classification process, and an area-adjusted accuracy assessment was conducted through cross-validation and test datasets. The results demonstrate that the country-wide forest stand species mapping achieved an accuracy exceeding 80%, however it varies greatly depending on

species, region and observation frequency. We provide freely accessible resources including the forest tree species map, training and test data: https://doi.org/10.5281/zenodo.10180469 (Grabska-Szwagrzyk, 2023).

## 1. Introduction

Information of forest tree species composition is essential for many scientific applications as well as the purposes of the forest inventory and management, such as estimating timber volume, modelling biodiversity, conservation, monitoring of

disturbances or carbon and biomass estimation (Hanewinkel et al., 2013; Loiselle et al., 2003; Gillis et al., 2005; Boisvenue and White, 2019). In recent times, use of remote sensing data has greatly improved forest monitoring and management. One such powerful source of data is Sentinel-2 mission, which offers high-resolution and frequent data for mapping tree species. While Sentinel-2 data have been increasingly employed for mapping species composition, most studies focus on smaller regional scales (Immitzer et al., 2016; Puletti et al., 2017; Karasiak et al., 2017; Persson et al., 2018; Grabska et al., 2019;

Immitzer et al., 2019; Hościło and Lewandowska, 2019; Bolyn et al., 2018; Grabska et al., 2020; Lechner et al., 2022; Shirazinejad et al., 2022; Axelsson et al., 2021; Wessel et al., 2018; Melnyk et al., 2023) or classify broad forest classes/species





groups over larger regions (Waser et al., 2021; Breidenbach et al., 2021; Schindler et al., 2021; Rüetschi et al., 2021). For larger areas, discrimination of tree species has been performed with the use of Landsat (Turlej et al., 2022; Bonannella et al., 2022). Furthermore, continent-scale studies have utilized high-resolution hyperspectral and field data to develop models for
tree species classification, evaluating both general and site-specific models (Marconi et al., 2022). At the national scale, Sentinel-2 time series were successfully used to map seven dominant tree species in Germany (Welle et al., 2022) or map larch plantations in Wales (Punalekar et al., 2021). In studying tree species composition for larger regions, additional environmental variables, for instance topographic predictors, have been found to improve classification accuracy (Waser et al., 2021; Grabska et al., 2020; Ye et al., 2021). Other datasets used as auxiliary variables include climatic variables (Hermosilla et al., 2022),
phenological metrics (Kollert et al., 2021; Hermosilla et al., 2022), spectral indices (Schindler et al., 2021; Ye et al., 2021; Hemmerling et al., 2021; Praticò et al., 2021) and textural metrics (Ye et al., 2021; Hemmerling et al., 2021).

Still, the accurate mapping of forest tree species with remote sensing data remains a challenge (Fassnacht et al., 2023). Particularly, studying species composition in large areas presents significant problems, such as generating good quality predictors from satellite imagery (Grabska et al., 2020). The frequent cloud cover or topographic effects in mountainous
regions may limit the number of cloud-free observations or disturb the surface reflectance values (Schindler et al., 2021). Additionally, larger areas exhibit greater environmental variability, including variations in topography, climate, and phenology, which can significantly impact species classification accuracy. The optimal image acquisition dates which are crucial in improved species recognition (Grabska et al., 2019; Immitzer et al., 2019) may substantially differ between regions. Other challenge in large-scale classification is the limited availability of reference data, especially for less common species
(Zeug et al., 2018), leading to poorer performance for underrepresented species (Hemmerling et al., 2021; Marconi et al., 2022; Ahlswede et al., 2022). Finally, species classification for large regions requires handling high-volume spatial datasets, which may be difficult to process using locally installed, monolithic software. Google Earth Engine (GEE), the - for research purposes - freely accessible cloud-based platform, enables parallel processing of large spatial datasets (Tamiminia et al., 2020; Gorelick et al., 2017). GEE provides the access to entire, pre-processed Sentinel-2 collections and other environmental datasets as well
as tools for processing and classification (Tamiminia et al., 2020). Previous studies have demonstrated the potential and versatility of GEE in forest classification, emphasizing its role in addressing the challenges encountered in large-scale mapping (Forstmaier et al., 2020; Chen et al., 2017; Praticò et al., 2021). Different approaches have been used to produce seamless and cloud-free satellite composites for mapping tree species composition, with multiple studies emphasizing the importance of a multi-temporal approach for accurate tree species classification (Immitzer et al., 2019; Grabska et al., 2019; Hościło and
Lewandowska, 2019; Persson et al., 2018; Kollert et al., 2021). However, there are variations in optimal timing for different seasons, and applying a single seamless image composition at a country-wide scale is not feasible. Thus, researchers often employ temporal aggregations such as spectral-temporal metrics (STM) calculated for a season, year, or multi-annual periods.



Here, we present classification of 16 forest tree species/genera for the entire area of Poland. Given the availability of several years of Sentinel-2 imagery, we propose, based on our previous findings (Grabska et al., 2020, 2019), a novel approach that

utilizes short-period (15-30 days) seasonal composites derived from multiple years. This strategy aims to focus on critical periods characterized by dynamic phenological changes while avoiding gaps in imagery that are commonly encountered when using single-year data. We used GEE for pre-processing and classification of the Sentinel-2 time series, along with additional environmental variables.

## 2. Data and methods

### 2.1 Study area

Poland's forests cover an area exceeding nine million hectares - 9,265,000 ha according to the Central Statistical Office; (December 31, 2021); or 9,464,000 ha according to the standard adopted for international assessments, taking into account land related to forest management (Zajączkowski et al., 2022). This accounts for approximately 30% of the country's total land area (Figure 1). In terms of ownership, public forests hold the majority share at 80.7% (with 76.9% of forests managed by the

State Forests, 2% belonging to National Parks, and 1.8% of communes' properties and others), followed by private forests at around 19.3%. The dominant species is the Scots pine (*Pinus sylvestris*), covering 58.5% of the forested area across all ownership types, according to the National Forest Inventory (NFI) reports (Biuro Urządzania Lasu i Geodezji Leśnej, 2022). The second most prevalent genus is *Quercus*, primarily *Robur* and *Pedunculate* species, accounting for 8.0%. Birch (*Betula pendula*) represents 6.8% and alder species (*Alnus* spp.) 5.7% of tree species. In the mountainous regions in southern Poland,

Norway spruce (*Picea abies*), silver fir (*Abies alba*), and common beech (*Fagus sylvatica*) are the most common species, covering 5.3%, 3.3%, and 6.2%, respectively. It is worth noting that European larch (*Larix decidua*) shares are usually not

reported separately but in combination with pine species. Still, larch is also among prevalent species in Poland - based on data from the Polish Forest Data Bank (FDB), the share of larch in Poland's State Forests land property is approximately 2%.

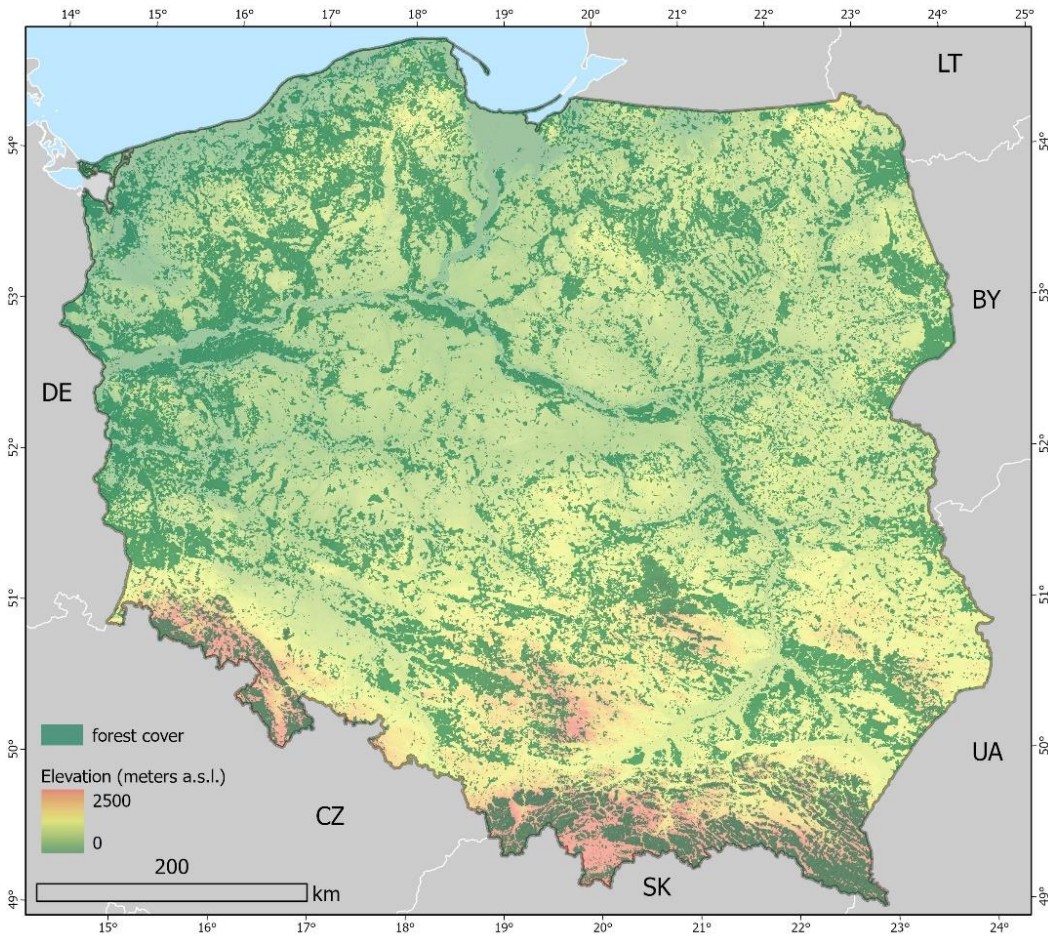

**Figure 1** Forest cover in Poland Elevation from EU-DEM; forest mask derived in this study.

## 2.2. Workflow

We developed an approach to classify 16 tree species in Poland using Sentinel-2 time series within the GEE platform. Polish FDB was used as reference data for training, validation and test samples. We created four seasonal composites (averages) from multi-annual observations (2018-2021), performed pre-processing in GEE, and clipped the final composites to the forest mask derived from existing land cover datasets. Classification involved the Random Forest (RF) classifier with a 10-fold cross-validation technique, and accuracy metrics were computed using test samples. To handle class imbalances, we implemented two strategies: proportional and unproportional allocation. Additionally, we compared accuracy between areas influenced by overlapping and non-overlapping Sentinel-2 orbits.



### 2.3. Reference data processing

The reference data was gathered from publicly accessible FDB, in which forest management units (forest stands) are represented by polygons. Each polygon contains information on species share expressed by values ranging from 1 to 10, with ten indicating homogenous coverage by a particular species. Nonetheless, the precise spatial distribution of these species within the polygons remains uncertain. In addition, the FDB does not cover private forests.

From FDB, polygons representing pure stands with a single species dominance of 90% or more, and with trees older than 10
years were selected. However, due to frequency of some species in Polish forests, we used other thresholds and additional conditions. Given the large number of reference stands of Scots pine, we randomly chose 10% of pure stands with 100% share of this species, however other pine species uncommon in Poland underwent the same processing procedures as other examined species. On the other hand, due to an insufficient number of reference samples for less common species such as poplar (*Populus* spp.), black locust (*Robinia pseudoacacia*), hornbeam (*Carpinus betulus)*, ash (*Fraxinus excelsior*), maple (*Acer* spp.) and
Douglas fir (*Pseudotsuga menziesii*), additional FDB stands with a 60-80% share of these species were included. The next step involved precise adjustments of reference samples to the actual forest mask derived from two available land cover datasets in GEE, i.e. any samples or their parts falling outside of forest mask were removed. Specifically, we utilized the ESA WorldCover 2021 product ("ESA/WorldCover/v200"; (Zanaga et al., 2022)), selecting only value 10 (i.e., tree cover), and the Dynamic World dataset ("GOOGLE/DYNAMICWORLD/V1", (Brown et al., 2022)) calculated from summer 2021 imagery and
aggregated to mean, with a tree probability threshold set at 0.6. Both datasets were employed, as based on our tests, the ESA World Cover product tends to overestimate forests in certain areas, while the DynamicWorld dataset, generated dynamically from available Sentinel-2 observations, may be prone to errors due to frequent cloud cover. In the next step, image segmentation on the Sentinel-2 composite was performed (Harmonized Level-2A data; 'COPERNICUS/S2_SR_HARMONIZED'), utilizing mean values from summer 2021. This segmentation process was carried
out using Simple Non-Iterative Clustering (SNIC; (Achanta and Süsstrunk, 2017)) algorithm in GEE limited to the previously selected FDB stands within the forest mask area, with the aim to delineate spectrally homogeneous patches. Segments obtained in this step were intersected with the FDB stands, and for further processing only segments larger than 0.5 hectares that encompassed more than 60% of the stands were selected. Subsequently, the resulting segments were visually checked using very high-resolution orthoimagery. Finally, 4500 polygons were obtained representing 16 species/genera (Table 1). They were
divided into training (2999; corresponding to approx. 400 thousands training pixels) and test polygons (1501). The training data was further divided into training (90%) and validation (10%) and 10-fold cross-validation was employed to calibrate the model. The examples of reference samples for each examined class are illustrated in Figure 2.

**Table 1** Classes and species classified in our study

| Class | Species | No. of polygons |
|-------|---------|-----------------|
| **Pine** | *Pinus sylvestris* | 1036 |
|  | *Pinus nigra* |  |

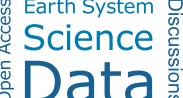

|  | | |
|---|---|---|
|  | *Pinus strobus* | |
|  | *Pinus rigida* | |
|  | *Pinus banksiana* | |
| **Oak** | *Quercus robur* | 512 |
|  | *Quercus petraea* | |
|  | *Quercus rubro* | |
| **Beech** | *Fagus sylvatica* | 301 |
| **Alder** | *Alnus glutinosa* | 477 |
|  | *Alnus incana* | |
| **Birch** | *Betula pendula* | 419 |
|  | *Betula pubescens* | |
| **Larch** | *Larix Decidua* | 256 |
| **Spruce** | *Picea abies* | 419 |
| **Fir** | *Abies alba* | 171 |
| **Hornbeam** | *Carpinus betulus* | 134 |
| **Poplar** | *Populus alba* | 176 |
|  | *Populus tremula* | |
|  | *Populus nigra* | |
| **Ash** | *Fraxinus Excelsior* | 164 |
| **Maple** | *Acer pseudoplatanus* | 122 |
|  | *Acer platanoides* | |
|  | *Acer campestre* | |
| **Lime** | *Tillia cordata* | 60 |
|  | *Tilia platyphyllos* | |
| **Douglas fir** | *Pseudotsuga menziesii* | 124 |
| **Black locust** | *Robinia pseudoacacia* | 86 |
| **Dwarf mountain pine** | *Pinus mugo* | 43 |

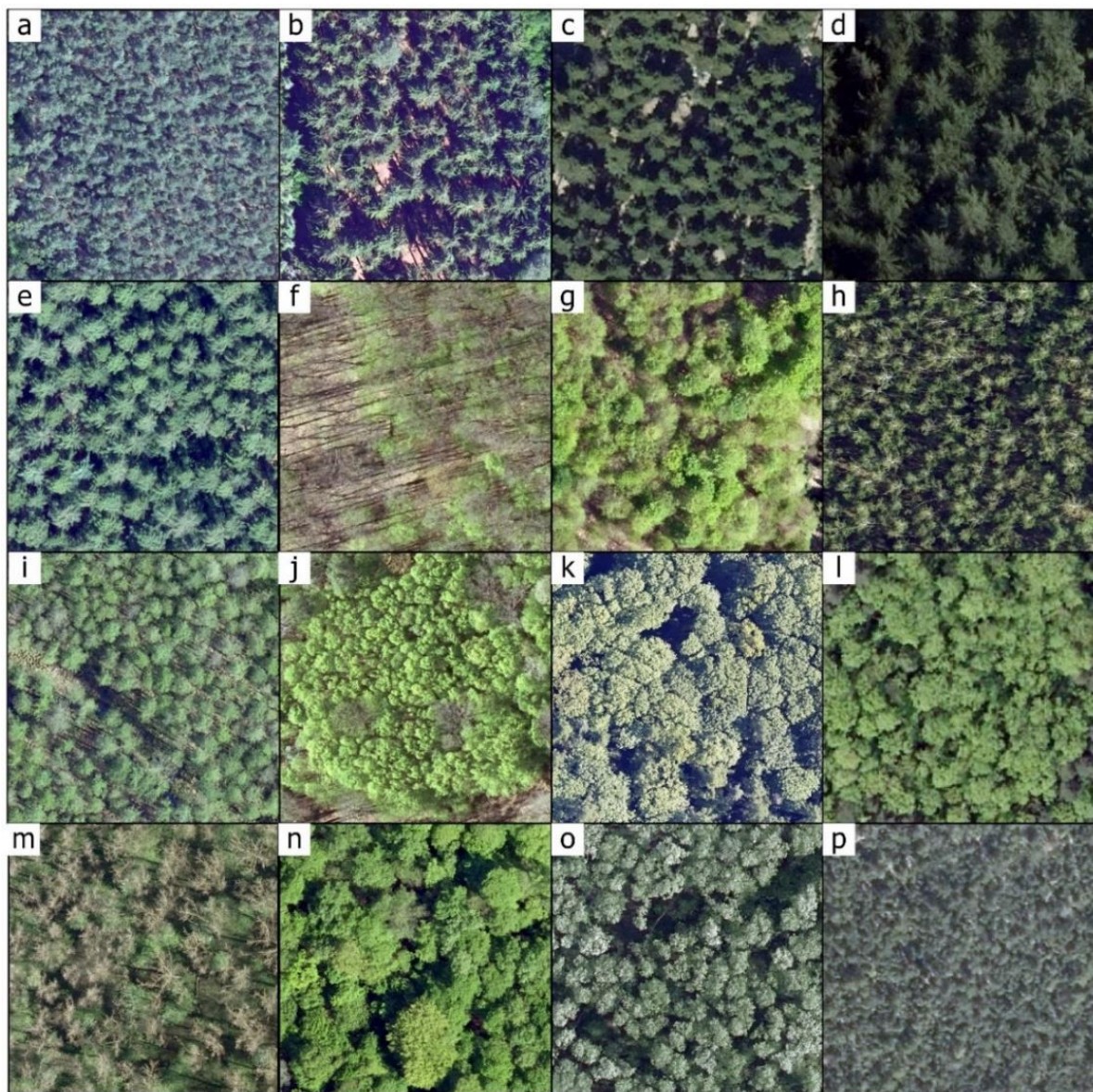

**Figure 2** Examples of reference samples for each analysed tree species/genera shown in (spring) very high resolution orthoimagery: a) pine; b) spruce; c) fir; d) Douglas fir; e) larch; f) oak; g) beech; h) birch; i) alder; j) hornbeam; k) maple ; l) ash; m) poplar; n) lime; o) black locust; p) dwarf mountain pine. Orthoimagery is openly available at Polish Geoportal (https://mapy.geoportal.gov.pl/; Head Office of Geodesy and Cartography).

## 2.4. Satellite imagery processing and additional variables

Regarding satellite imagery composites, numerous studies have demonstrated the significance of a multi-temporal approach in accurately distinguishing tree species (Immitzer et al., 2019; Grabska et al., 2019; Hościło and Lewandowska, 2019; Persson et al., 2018; Kollert et al., 2021). For instance, our previous study on species classification in a smaller area highlighted the



optimal timing for distinguishing forests tree species in temperate zones, which varies during the spring and autumn seasons
(Grabska et al., 2019). At a national scale, however, applying a single seamless image composition for the entire growing season is impractical. While seasonal STMs can provide important phenological information (Müller et al., 2015), areas with frequent cloud-cover may still experience difficulties in acquiring high-quality observations for all needed temporal time steps (Grabska et al., 2020). Different approaches to calculate Sentinel-2 based STMs were employed, such as utilizing seasonal metrics calculated over two to four months (Praticò et al., 2021) or testing long-term, seasonal, and monthly composites (Nasiri
et al., 2023).

Here, we employed seasonal Sentinel-2 (L2A) Spectral-Temporal Metrics (STM) calculated in GEE for four periods: (1) the second half of April, (2) May, (3) June/July, and (4) October from the years 2018-2021. For each period, one seasonal composite from multi-annual observations was calculated. The specific periods for each season and year are provided in Table 2. These composites were selected based on findings from our previous studies (Grabska et al., 2019, 2020; Grabska-
Szwagrzyk and Tymińska-Czabańska, 2023). The spring imagery was chosen to capture the greening-up phase, while autumn imagery was selected to represent the period when leaves undergo colour changes. Furthermore, we decided to include two spring composites, one early and one late spring, as our previous study revealed significant differences among deciduous species in this period. For instance, on a smaller site, there was an 8-18 day gap between early leafing species like larch and birch, and late leafing species like alder and oak (Grabska-Szwagrzyk and Tymińska-Czabańska, 2023). Moreover, we
included a summer composite, as it represents a relatively stable and certain period and allows to utilize a greater number of images. In the previous study on forest tree species classification in the Polish Carpathians, bands from July composite were among the most important variables (Grabska et al., 2020). The dates were slightly modified due to meteorological conditions in particular years and therefore phenology variations, as well as missing observations in some cases (Table 2).

All available Sentinel-2 images from the Harmonized Level-2A collection captured during these periods and with cloud cover
below 40% were pre-processed, including cloud, cloud shadow and dark pixel masking based on the Sentinel-2 cloud probability dataset (based on the Sentinel2-cloud detector, see https://github.com/sentinel-hub/sentinel2-cloud-detector) also available in GEE ('COPERNICUS/S2_CLOUD_PROBABILITY'). The number of clear observations for each period varied largely due to cloud cover as well as overlapping Sentinel-2 orbits (Figure 3).

**Table 2 Periods of Sentinel-2 imagery used in analysis.**

| Name | 2018 | 2019 | 2020 | 2021 |
|---|---|---|---|---|
| **Early spring** | 04/15 – 05/10* | 04/20 – 05/10 | 04/20 – 05/10 | 05/05 – 05.25 |
| **Late spring** | 05/10 – 05/30 | 05/15 – 06/05 | 05/15 - 06/05 | 05/25 – 06/15 |
| **Summer** | 06/10 – 07/10 | 06/10 – 07/10 | 06/10 – 07/10 | 06/10 – 07/10 |
| **Autumn** | 10/25 – 11/10 | 10/20 – 11/05 | 10/20 – 11/05 | 10/25 – 11/10 |



*Period increased due to not enough observations

The pre-processed imagery was then clipped to match the actual forest mask, ensuring that only relevant areas were considered for analysis. In addition, the Normalized Difference Vegetation Index (NDVI) was calculated to mitigate the potential impact of disturbances on the obtained results and remove recent clear cuts, ensuring that only areas with healthy vegetation were considered. Specifically, based on tests, the pixels with NDVI values below 0.6 from the summer 2021 composite were
excluded from the analysis (Figure 4). Final step employed calculating average reflectance values for each pixel, and for each specific season, based on the seamless Sentinel-2 imagery.

Additional variables for classification included environmental datasets available in GEE. They included: elevation data (reprocessed 30m SRTM data: "NASA/NASADEM_HGT/001"), WorldClim variables ("WORLDCLIM/V1/BIO"): temperature and precipitation (bio1, bio12, bio17), soils ("OpenLandMap/SOL/SOL_GRTGROUP_USDA-
SOILTAX_C/v01") and Terra Climate ('IDAHO_EPSCOR/TERRACLIMATE') maximum air temperature for 2018.

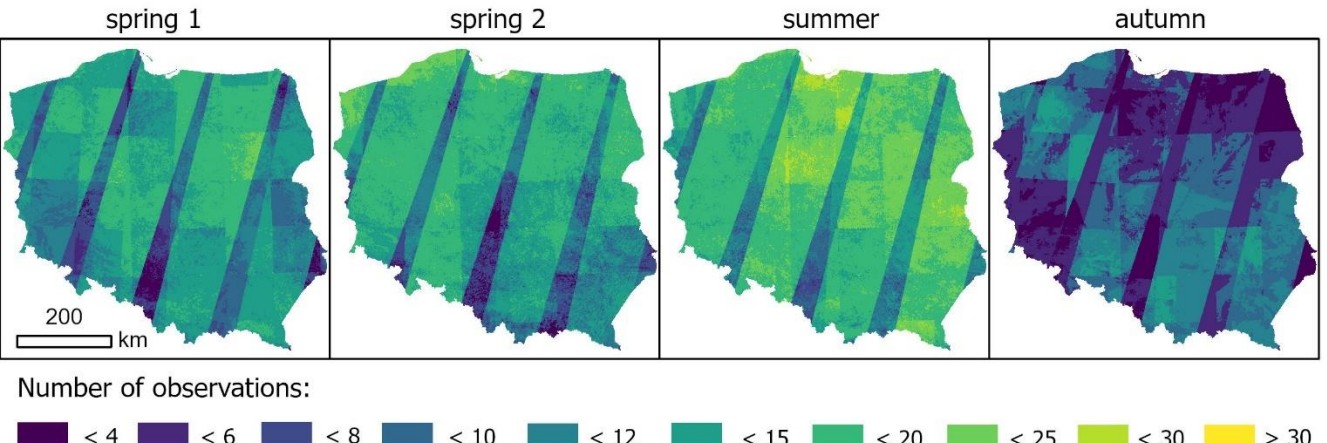

**Figure 3** Number of cloud-free observations in the analysed periods combined for all years.

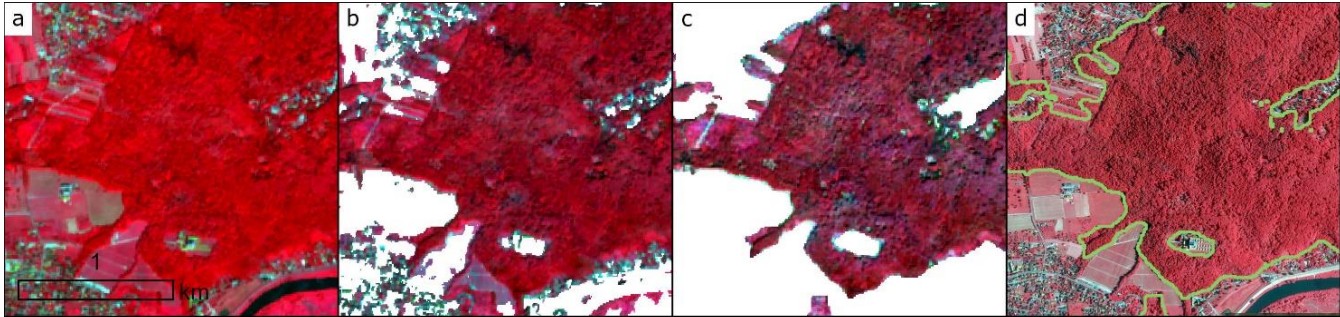

**Figure 4** Procedure for obtaining forest mask used in this study on the example of part of Kraków, southern Poland: a) Sentinel-2 image
(NIR, VIS R, VIS G); b) Sentinel-2 clipped to ESA World Cover v200 dataset extracted tree cover class; 3) Sentinel-2 clipped to ESA World Cover, Dynamic World and NDVI thresholding; 4) high resolution CIR orthoimage with borders of the calculated forest mask.

## 2.5. Classification and accuracy assessment.





Classification for the entire area of Poland was performed using approx. 400,000 sample pixels, employing a 10-fold cross-validation technique. RF classifier (Breiman, 2001) was used within the GEE, with the number of trees set to 200. Accuracy
assessment included estimation of area-adjusted confusion matrices, producer's accuracy (PA), user's accuracy (UA), F1-score, which is a weighted harmonic mean of UA and PA, and overall accuracy (OA). For this task, 1501 test polygons (see section 2.3) were utilized. To ensure the robustness of accuracy assessment, a stratified random sampling approach based on species was adopted, as recommended by Olofsson et al. (2014) and based on our previous research (Grabska et al., 2020). Furthermore, we tested the unproportional allocation approach which is commonly employed when dealing with substantial
class imbalances (Marconi et al., 2022; Maxwell et al., 2018; Jackson and Adam, 2021).

In recognition of class imbalance, a two-fold strategy was implemented. The first approach involved proportional allocation, while the second approach involved an unproportional dataset. The sample size for less common species was increased through oversampling, whereas undersampling was employed to the most common class, *Pinus.* In both approaches, the size of the sample was approximately 20,000 pixels (see the detailed table in the appendix) and a minimum sampling distance of 20
meters was used. Finally, regarding the significant differences in number of observations between Sentinel-2 orbit overlapping and non-overlapping areas, further analyses were conducted to evaluate the impact of observation frequency on accuracy. This included the calculation of OA separately for overlapping and non-overlapping areas in both sampling approaches.

### 3. Results and discussion

#### 3.1. Overall accuracy of the tree species maps

On average, the classification process yielded high OA, achieving values of approximately 80% or higher. Employing a 10-fold cross-validation, the average OA was equal to 83.3%, ranging between 79.3 and 84.9%. Subsequently, the species map with the best performance in terms of OA from the initial step was validated with approximately 20,000 pixels in two approaches: proportional and unproportional. The proportional approach demonstrated an OA of 89.6%, while in the unproportional approach a lower accuracy of 84% was achieved. This decline in accuracy when transitioning from proportional
to unproportional samples allocation is reasonable, as more samples represent less-common species, usually underperforming the most common ones.

OA varied between regions with overlapping and non-overlapping Sentinel-2 orbits. The following OAs were obtained: 86.7% for not-overlapping areas and 90.1% for overlapping area using proportional allocation, and 83.8% and 84.1% using unproportional allocation respectively. Although the difference using unproportional allocation seems to be low, limited
number of clear observations may increase the uncertainty of estimations (Schindler et al., 2021). In studies which utilize Landsat imagery, the number of clear observations plays a vital role in classification accuracy improvement (Turlej et al., 2022). Furthermore, in mapping large areas accuracy metrics are not expected to be uniform in space due to high species and environmental diversity. Examples of selected regions with low and high accuracies are illustrated in Figure **5**. Numerous





environmental and forest-related factors can impact the results. For example, heterogeneous forest structure with high diversity
in age and species (Figure **5**A) result in misclassifications and require further examination and addressing. Also, misclassification occurs more often in the mountainous areas, particularly in the Carpathian forests due to higher species and environmental diversity and topography effects (Figure **5**B). High accuracy is observed in areas featuring a combination of various species but composing pure stands with a similar forest structure (Figure **5**C) as well as in locations where dense black locust stands are present (Figure **5**D).

**3.2. Tree species distribution and accuracy**
The obtained map of forest tree species/genera reveals the share and spatial distribution of forests in Poland. Pine-dominated stands are the most common, accounting for 47.5% of the total forest cover in the country. Several other common species prevalent across Polish forests are birch occupying 11.7% of the forested areas, along with alder at 9%, beech at 8.1%, and oak at 7.2%. Other common species include spruce (3.7%) and fir (2.8%), predominantly occurring in mountainous areas in
the southern Poland. Additionally, larch-dominated stands are relatively common (3.6%), along with ash (1.7%), hornbeam (1.1%) and poplar (1%). Several other species each hold a share of less than 1% in the overall forest composition, including Douglas fir, maple, and black locust. Lastly, lime and dwarf mountain pine have a more marginal presence in the obtained map.

The comparison of the results with official statistics shows some discrepancies. Firstly, the share of pine in our map is
underestimated by more than 10 percentage points, which may result from several factors. One possible reason is the misclassification of pine as spruce or other coniferous trees, which accounts for 0.65% of the reference data, particularly in mountainous regions. Additionally, the share of pine is decreasing in recent years due to shifts in forest management practices, such as the transition from monocultures to stands with more diversified species composition (Tomaś and Jagodziński, 2019). Furthermore, pine has been susceptible to disturbances in recent years, which may have led to misclassifications (Hemmerling
et al., 2021). Another species with share lower in our map than reported is spruce (3.6% vs 5.3%), which, in recent years is exposed to significant disturbances and dieback, particularly in the Western Carpathian mountains and Białowieża forest (Grodzki, 2010; Bałazy, 2020; Kamińska et al., 2021). Consequently, the share of spruce is also decreasing. On the other hand, certain species like alder and birch are seemingly more common than in the official reports. The larger share of birch may be attributed to the fact that this species are common on abandoned agricultural land, it is also regarded as pioneer and successional
species (Hynynen et al., 2010). The area analysed in our study might include former agricultural lands where forest succession takes place, a process that is very common in different parts of Poland (Shahbandeh et al., 2022; Kolecka et al., 2017; Zgłobicki et al., 2020; Majchrowska, 2013). Abandoned areas with forest succession, however, are not included in the official reporting for forests. Also, while very young forests have been excluded from our analysis, the visual inspection indicates frequent misclassifications of younger stands covered with broad-leaved trees as alder, which may be one of the reasons for its
overestimation.



**Figure** 5 Examples of classification (middle) compared with high-resolution orthoimagery (left) and dominant species from Forest Data Bank (right): a) Czarna Białostocka forest district, NE Poland lowlands; b) Baligród forest district; SE Poland, Bieszczady mountains; c)



Kłodawa forest district, NW Poland lowlands; d) Sulechów forest district; W Poland lowlands. Orthoimagery is openly available at Polish Geoportal (https://mapy.geoportal.gov.pl/; Head Office of Geodesy and Cartography).

In terms of species accuracy, the most abundant species in Poland, pine, was classified with the highest accuracy, exceeding 90% F1-score (Figure 6). Other species demonstrating F1-score of 80% or higher included dwarf mountain pine, alder, beech, fir, spruce, oak and larch. With the exception of dwarf pine mountain, these species are common in forests of Poland. On the other hand, the classification of poplar, Douglas fir, maple, lime, hornbeam, and ash revealed relatively poor accuracy levels,

below 60%. Surprisingly, rare species such as black locust achieved high classification accuracy around 75%. Confusion matrix reveals the frequent misclassifications (Table 3). Typically, broad-leaved species such as ash, hornbeam and lime are misclassified – ash and lime as oak and hornbeam as oak and beech species; while coniferous Douglas fir – as pine. Similarly, in the study of Hemmerling et al. (2021), less common species with relatively high accuracy was black locust. This is a result of its unique spectral-temporal properties, as usually it leaves out later than other broad-leaved and is characterized by

flowering in late spring (Rusňák et al., 2022; Somodi et al., 2012). It is promising result, taking into account the invasiveness of this non-native species in Europe (Richardson and Rejmánek, 2011). The visual inspection also indicates that frequent misclassifications include younger stands, such as oak, misclassified as other broad-leaved species, e.g. alder. Importantly, the age structure within the examined species differs largely (based on FDB), with average values between approx. 50 years old for birch, larch and alder; around 70 for spruce and pine and above 80 for beech, oak and fir. Furthermore, the species

classification in young forests, characterized by the distinguished spectral characteristics than the mature ones, is challenging. Finally, not all species occurring in Poland were classified.

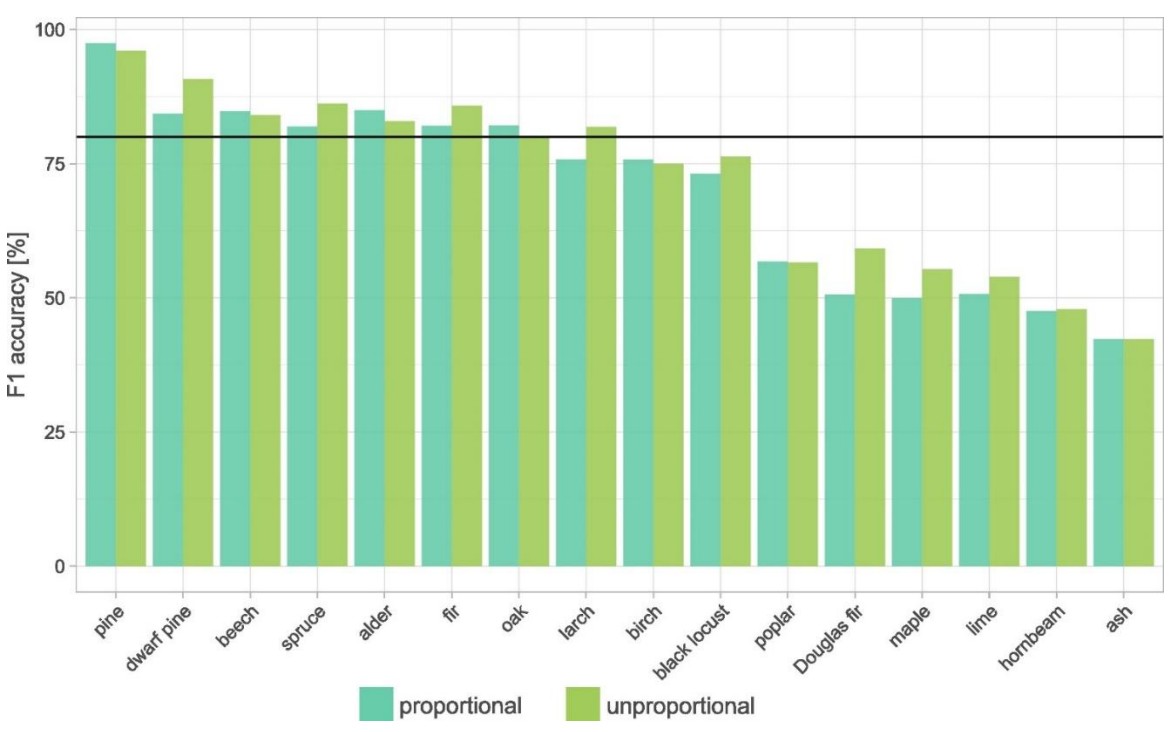





**Figure 6** F1-score for 16 analysed species in two approaches: proportional samples allocation and unproportional allocation with down sampling of pine and oversampling of other classes.

**Table 3** Area-adjusted confusion matrix for the unproportional samples allocation (populated by estimated proportions of area).

| | 1 | 2 | 3 | 4 | 5 | 6 | 7 | 8 | 9 | 10 | 11 | 12 | 13 | 14 | 15 | 16 |
|---|---|---|---|---|---|---|---|---|---|---|---|---|---|---|---|---|
| **Beech (1)** | 7.51 | 0.12 | 0.28 | 0.00 | 0.46 | 0.01 | 0.08 | 0.13 | 0.02 | 0.01 | 0.17 | 0.01 | 0.08 | 0.01 | 0.02 | 0.00 |
| **Birch (2)** | 0.07 | 7.14 | 0.35 | 0.02 | 0.07 | 0.02 | 0.01 | 0.01 | 0.02 | 0.21 | 0.29 | 0.05 | 0.34 | 0.01 | 0.01 | 0.00 |
| **Oak (3)** | 0.85 | 1.16 | 10.77 | 0.00 | 0.81 | 0.01 | 0.21 | 0.16 | 0.27 | 0.13 | 0.21 | 0.00 | 0.40 | 0.00 | 0.10 | 0.00 |
| **Douglas f. (4)** | 0.00 | 0.01 | 0.00 | 0.61 | 0.00 | 0.25 | 0.01 | 0.00 | 0.00 | 0.01 | 0.01 | 0.12 | 0.00 | 0.04 | 0.00 | 0.00 |
| **Hornbeam (5)** | 0.27 | 0.03 | 0.10 | 0.00 | 1.12 | 0.00 | 0.03 | 0.14 | 0.15 | 0.00 | 0.09 | 0.00 | 0.02 | 0.00 | 0.00 | 0.00 |
| **Fir (6)** | 0.03 | 0.00 | 0.00 | 0.08 | 0.01 | 5.50 | 0.00 | 0.00 | 0.00 | 0.01 | 0.01 | 0.13 | 0.00 | 0.18 | 0.00 | 0.00 |
| **Ash (7)** | 0.03 | 0.03 | 0.06 | 0.00 | 0.06 | 0.00 | 0.46 | 0.07 | 0.08 | 0.00 | 0.34 | 0.00 | 0.02 | 0.00 | 0.04 | 0.00 |
| **Maple (8)** | 0.03 | 0.00 | 0.01 | 0.00 | 0.01 | 0.00 | 0.01 | 0.42 | 0.01 | 0.00 | 0.05 | 0.00 | 0.01 | 0.01 | 0.00 | 0.00 |
| **Lime (9)** | 0.00 | 0.00 | 0.01 | 0.00 | 0.02 | 0.00 | 0.01 | 0.01 | 0.39 | 0.00 | 0.00 | 0.00 | 0.00 | 0.00 | 0.00 | 0.00 |
| **Larch (10)** | 0.03 | 0.54 | 0.02 | 0.01 | 0.01 | 0.07 | 0.00 | 0.00 | 0.01 | 3.57 | 0.03 | 0.16 | 0.04 | 0.05 | 0.00 | 0.00 |
| **Alder (11)** | 0.15 | 0.90 | 0.13 | 0.00 | 0.10 | 0.01 | 0.15 | 0.03 | 0.03 | 0.01 | 7.29 | 0.04 | 0.06 | 0.03 | 0.08 | 0.00 |
| **Pine (12)** | 0.00 | 0.28 | 0.01 | 0.22 | 0.00 | 0.14 | 0.00 | 0.00 | 0.00 | 0.20 | 0.03 | 29.56 | 0.02 | 0.44 | 0.02 | 0.06 |
| **Poplar (13)** | 0.00 | 0.16 | 0.06 | 0.00 | 0.05 | 0.00 | 0.02 | 0.01 | 0.06 | 0.01 | 0.02 | 0.00 | 0.91 | 0.00 | 0.00 | 0.00 |
| **Spruce (14)** | 0.01 | 0.00 | 0.00 | 0.08 | 0.00 | 0.90 | 0.00 | 0.00 | 0.00 | 0.03 | 0.00 | 0.51 | 0.00 | 7.43 | 0.00 | 0.08 |
| **Black l. (15)** | 0.00 | 0.07 | 0.03 | 0.00 | 0.00 | 0.00 | 0.00 | 0.00 | 0.00 | 0.00 | 0.03 | 0.00 | 0.01 | 0.00 | 0.66 | 0.00 |
| **Dwarf p. (16)** | 0.00 | 0.00 | 0.00 | 0.00 | 0.00 | 0.00 | 0.00 | 0.00 | 0.00 | 0.00 | 0.00 | 0.01 | 0.00 | 0.00 | 0.00 | 0.68 |
| **Prop (ref)** | 8.98 | 10.42 | 11.82 | 1.02 | 2.70 | 6.89 | 0.97 | 0.97 | 1.03 | 4.19 | 8.58 | 30.57 | 1.91 | 8.20 | 0.93 | 0.82 |
| **Prop (map)** | 8.88 | 8.60 | 15.09 | 1.05 | 1.96 | 5.93 | 1.18 | 0.54 | 0.43 | 4.53 | 9.01 | 30.97 | 1.29 | 9.04 | 0.80 | 0.69 |
| **PA** | 83.6 | 68.5 | 91.1 | 60.3 | 41.4 | 79.9 | 47.0 | 43.0 | 38.3 | 85.3 | 85.0 | 96.7 | 47.4 | 90.6 | 70.8 | 83.6 |
| **UA** | 84.6 | 83.0 | 71.4 | 58.2 | 57.0 | 92.8 | 38.5 | 77.7 | 91.5 | 78.8 | 80.9 | 95.4 | 70.3 | 82.2 | 82.9 | 99.2 |

## 3.3 Limitations in large-area species mapping and proposed solutions

In the country-wide or other large-extent mapping cases, there are several challenges and limitations. Larger regions are often characterized by higher diversity of species and environmental conditions. Certain species occur only in spatially limited areas

– for example, in Poland, Silver fir is typical for the mountain areas only, while oaks and hornbeams tend to occur more often in the lowlands. In addition, due to the variability in meteorological conditions, the optimal period for classification of specific species may differ largely among regions, particularly during the spring when processes of leaf unfolding take place, and autumn while leaf coloring occur. Furthermore, these optimal periods may vary from year-to-year due to variations in spring temperatures and other meteorological conditions (Grabska-Szwagrzyk and Tymińska-Czabańska, 2023). Future research should also consider specific periods of imagery acquisition when aimed to distinguish different species, i.e. covering periods when particular species exhibit the highest phenological variations. It would be profitable to use multiple autumn (e.g. early and late autumn) composites, however it is very challenging due to insufficient number of clear observations during this time of the year.

One solution may be the division of the study area into smaller regions – in the country-wide or other large-extent mapping of species composition, the subdivision to smaller parts may play an important role, also due to computational power; similarly as in the study from Pazúr et al. (2022) or Hermosilla et al. (2022). However, another question arises how to define the optimal borders of smaller regions to achieve higher accuracy of the obtained map, which is rarely discussed in studies focused on remote sensing-based classification.

Another methodological challenge is the underrepresentation of clear observations in some regions. In this study, we employed short-period seasonal composites from Sentinel-2 time series rather than one seasonal average, as the information from specific periods of growing season is crucial in distinguishing species. In calculation of seasonal averages, multi-annual observations were used, still, for some regions the underrepresentation of clear observations occurs. It may have significant impact on map accuracy in regions of lower observation frequency. In the case of Poland, it is particularly observed in the places where two orbits do not overlap, specifically for autumn (Figure 3). This issue should be addressed in studies on species classification for larger regions using Sentinel-2 or similar satellite constellations.

As a result of abovementioned factors, the design of robust training, test and validation datasets is challenging. Finally, in certain regions such as privately-owned forests or lands not officially reported as forests (e.g., successional forests that have emerged on previously abandoned agricultural lands), there is no reference data available. These areas tend to exhibit greater complexity, making the task of assessing classification accuracy particularly demanding.

**4. Conclusions**

We have obtained the first national-scale forest tree species map for Poland, achieving an accuracy exceeding 80%. This was accomplished through a novel approach that involved the calculation of Sentinel-2 seasonal composites spanning multiple years. The resulting map is an important dataset for both forest management and the scientific community, facilitating tasks like modeling biodiversity and monitoring non-native and invasive species. It can enhance our understanding of forest ecosystems and support more informed and precise forestry and conservation effort. Unlike other existing data sources, such





as the FDB, which primarily provide information about the share of species within forest stands, this new map offers a view of tree species distribution at a finer scale. Furthermore, our map provides a unique advantage over traditional forest inventories like NFI, which offers point-based data rather than continuous spatial representation of species distribution.

## 5. Data availability

We provide freely accessible resources including the forest tree species map, training and validation data:
https://doi.org/10.5281/zenodo.10180469 (Grabska-Szwagrzyk, 2023). The map can be explored online: https://ee-aweaksbarg.projects.earthengine.app/view/speciesmappl

## Competing interests

The contact author has declared that none of the authors has any competing interests.

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



## Appendix

**Table A1: Number of test pixels for accuracy assessment in two approaches – proportional and not proportional.**

|                       | Estimated proportions | Proportional | Not proportional |
|-----------------------|-----------------------|--------------|------------------|
| **Pinus**             | 59%                   | 11,800       | 5900             |
| **Quercus**           | 8%                    | 1600         | 2400             |
| **Betula**            | 6.8%                  | 1360         | 2040             |
| **Fagus**             | 6.2%                  | 1240         | 1860             |
| **Alnus**             | 5.7%                  | 1140         | 1710             |
| **Picea**             | 5.3%                  | 1060         | 1600             |
| **Abies**             | 3.3%                  | 660          | 1320             |
| **Larix**             | 2%                    | 400          | 800              |
| **Carpinus**          | 1.3%                  | 260          | 520              |
| **Populus**           | 1%                    | 200          | 400              |
| **Fraxinus**          | <1%                   | 100          | 200              |
| **Pseudotsuga**       | <1%                   | 100          | 200              |
| **Acer**              | <1%                   | 100          | 200              |
| **Robinia pseudoacaccia** | <1%               | 100          | 200              |
| **Tilia**             | <1%                   | 100          | 200              |
| **Pinus mugo**        | <1%                   | 100          | 200              |
