# Peer review of "Map of forest tree species for Poland based on Sentinel-2 data"

_Earth System Science Data, 2023_

## Author Comment (AC1)

Dear Oleksandr Melnyk,

Thank you very much for your review. Based on your comments, we have provided justification for selecting the Random Forest algorithm in the text. Additionally, we have clarified the issue of training data, as well as the possibility of dividing the area into smaller regions.

Please see the responses to your comments in the table below.

| | |
|---|---|
| In general, the article made a pleasant impression, is well written and fully represents the methodology and results of the study. | Thank you very much for your kind words. |
| In this kind of research, the most important challenge is to create training samples. Based on open FDB data and to improve classification accuracy, it is worth conducting field validation, but on the scale of even regions, this task is very difficult and time-consuming. On the other hand, the frequency of FDB data updates is important. The classifier's accuracy may be impaired by deforestation that is overgrown with fast-growing vegetation in a year or two, which we have encountered in our research. | We agree with the reviewer that the issue of creating reliable training samples is an extremely important aspect in the classification of remote sensing data. Indeed, conducting field validation would be beneficial, but it is extremely time- and labour-consuming, particularly when the study area is large. Therefore, the reference samples from the FDB were thoroughly validated, both automatically and visually (see lines 100-119). We also acknowledge the challenges related to reference data (see lines 50-52 in introduction and 304-306 in the discussion).

Regarding FDB updates, this database is based on forest management plans created every 10 years for each forest district. However, the descriptions and maps are updated annually, based on the management practices carried out during that year (e.g., harvests, reforestation, etc.). Despite this, we are aware that there may still be some errors or inaccuracies in the FDB data. Therefore, in addition to semi-automatic validation, we also ensured that clear-cut or highly disturbed forest pixels were not considered. We applied the actual forest mask based on ESA World Cover, Dynamic World, and NDVI values calculated for the summer of 2021. Consequently, we excluded any areas from the reference samples that are currently not covered by trees (please check lines 107-119). |
| It is not clear from the text of the article why the Random Forest classification algorithm was chosen. This algorithm is very popular among researchers, although there are other effective algorithms whose results would be interesting to compare, especially on a national scale. | Thank you for your comment. We didn't aim for a comparison of different algorithms in this paper, but tried to use the one of the most promising machine learning algorithms. Together with SVM, RF is one of the most popular and powerful machine learning algorithms used in remote sensing analysis (apart from deep learning) nowadays. RF has also been successfully used in other studies for mapping vegetation in large areas. We selected the RF classifier also because of recommendations from the literature in respect to its insensitivity to overfitting and outliers in training samples. For instance, see Belgiu, M., Drăguţ, L., 2016. Random forest in remote sensing: A review of applications and future directions. ISPRS Journal of Photogrammetry and Remote Sensing 114, 24–31. https://doi.org/10.1016/j.isprsjprs.2016.01.011: "[Rf] is less sensitive than other streamline machine learning classifiers to the quality of training samples and to overfitting, due to the large number of decision trees produced by randomly selecting a subset of training samples and a subset of variables for splitting at each tree node." The third reason was the computational performance for the large area and data within Google Earth Engine. We have added justification for using RF in lines 181-185. |

| | |
|---|---|
| Perhaps, to improve the quality, it is worth dividing the territory of Poland, for example, by geographical provinces, although such a division is rather arbitrary. | This is a very valuable suggestion. In fact, in the initial tests of this study, we intended to divide Poland into regions. However, another challenge arose related to determining the optimal divisions (e.g., ecological forest regions, soil-botanical regions). This led to further challenges, such as a general decrease in the number of reference samples and increased imbalance among them. An example of such erroneous results occurred when dividing the regions into two. This can be seen in the figures below. The upper figure depicts Scots pine misclassified as silver fir (dark blue color) in the Carpathian region, where there is an underrepresentation of Scots pine samples. However, when both regions were used, the Scots pine were correctly classified (bottom image, purple color).
[Figure]

[Figure]
 We acknowledge that division may be a good solution but it also leads to further challenges, as discussed in lines 292-296. |
| The suggestions I have made do not in any way affect the quality of the work, and therefore I recommend it for publication. | Thank you very much for appreciating our study. |

Dear Jan Hemmerling,

Thank you for your review and insightful comments. Based on your feedback, we clarified the methodology for calculating STMs. Additionally, we included a table containing all the predictors and variable importance chart in the appendix. We also added additional columns in the Table 1 with the percentage of samples derived from stands with 60-80% share for less common species, as well as the number of reference pixels.

Please see the responses below.

| | |
|---|---|
| this manuscript covers a topic of constant relevance, is clearly written and for the most part easy to follow. The results are also sensibly discussed and summarised. | Thank you very much. |
| However, there are still some questions that need to be addressed with regard to the methods. | Thank you for all your comments and suggestions. Please see the responses below. We hope that we have provided more clarity in the Methods section. |
| For the reader, it is not clear how the multi-year aggregated features were created; in some places STMs are mentioned (e.g. lines 136, 138, 141), in others composites (lines 142, 147, 150). If these are composites, it would be important to know which rule set was used to create them. | Thank you for your comment, indeed the different terminology was not used sufficiently clearly. In the study we calculated only average (mean) STM from all the clear observations during the specified periods. Therefore, following your comment, we removed the term "composite" as it may suggest that more complicated rules were used and replaced it with either STM or other suitable term. |
| In the case of temporal statistics, it would be important for traceability to know which metrics were included as features in the classification. A list/overview of all features included in the classification would also be desirable. | Thank you for your suggestion. The table containing all variables used as predictors has been included in the appendix. |
| This also applies to the additional explanatory variables (from line 167). Has the effect of these variables on the classification results been tested? | Thank you for the comment, we have included a chart with variable importance in the appendix and commented this aspect in lines 221-228. |
| The period of the time windows for the creation of the composites/stms seems to overlap, at least in 2021 (Table 2.). Why was the summer period not shifted back? Is it possible that the same information was received in both STMs/composites? | Indeed, there is an overlap for a period of 5 days. This is related to the unusually late spring of 2021 and our decision to maintain the same dates for summer STMs. Despite this slight overlap, the final values between these two STMs for 2021 differ, as the mean was calculated. |
| To compensate for the differences in the number of training data, additional training data was taken from less common tree species from stands with a 60-80% mixing ratio (line 105). What proportion of the total number of training pixel in these classes did this account for? I think this is definitely | Thank you for your comment. The percentage of final samples derived from stands with a 60-80% share of specific species has been added to Table 1, along with additional columns indicating the number of pixels and their corresponding percentages for each class individually. |

| | |
|---|---|
| relevant for the interpretation of the results. Perhaps this could be added to the appendix? | |
| In general, however, I think that once these aspects have been clarified, nothing stands in the way of publication and I look forward to receiving comments on my remarks. | Thank you! |